# Suppressing hydrogen peroxide generation to achieve oxygen-insensitivity of a [NiFe] hydrogenase in redox active films

Huaiguang Li[1], Ute Münchberg[2], Alaa A. Oughli[1], Darren Buesen[1], Wolfgang Lubitz[3], Erik Freier [2] & Nicolas Plumeré [1✉]

Redox-active films were proposed as protective matrices for preventing oxidative deactivation of oxygen-sensitive catalysts such as hydrogenases for their use in fuel cells. However, the theoretical models predict quasi-infinite protection from oxygen and the aerobic half-life for hydrogenase-catalyzed hydrogen oxidation within redox films lasts only about a day. Here, we employ operando confocal microscopy to elucidate the deactivation processes. The hydrogen peroxide generated from incomplete reduction of oxygen induces the decomposition of the redox matrix rather than deactivation of the biocatalyst. We show that efficient dismutation of hydrogen peroxide by iodide extends the aerobic half-life of the catalytic film containing an oxygen-sensitive [NiFe] hydrogenase to over one week, approaching the experimental anaerobic half-life. Altogether, our data support the theory that redox films make the hydrogenases immune against the direct deactivation by oxygen and highlight the importance of suppressing hydrogen peroxide production in order to reach complete protection from oxidative stress.

[1] Center for Electrochemical Sciences, Ruhr-Universität Bochum, Bochum, Germany. [2] Leibniz-Institut für Analytische Wissenschaften – ISAS – e.V, 44227 Dortmund, Germany. [3] Max-Planck-Institut für Chemische Energiekonversion, Stiftstrasse 34-36, 45470 Mülheim an der Ruhr, Germany. ✉email: nicolas.plumere@rub.de

Redox-active polymer films applied as electron-conducting matrices for the immobilization of biological and bio-inspired catalysts on electrodes provide a practical route for their use in processes such as energy conversion. Recent reports demonstrated the effective integration of enzymes such as hydrogenases[1–4], nitrogenase[5], formate dehydrogenase[6], and photosystems[7–11] as well as molecular catalysts[12–14] in redox matrices on electrode surfaces. Since many of these catalysts are sensitive to $O_2$ to the extent that they deactivate within seconds under aerobic conditions[15], the polymer films were further engineered as protection matrices to avoid oxidative deactivation[1–3,6,9,16]. Electrons are transferred within the matrix by a hopping mechanism through redox moieties[17] such as viologens, quinones[18], or metal complexes[9]. By taking advantage of electron mediators that feature $O_2$-reducing properties, $O_2$ can be depleted at the outer layers of the film so that the catalysts in the inner layer remain under anaerobic conditions (Fig. 1). Viologen-modified polymer films (Fig. 1a) are particularly suitable for this task due to their fast electron transfer and $O_2$ reduction kinetics[19–24]. Theoretical models predict that these redox films can protect the catalysts against $O_2$ quasi-infinitely[3,25]. However, experimental observations on electrodes coated with hydrogenase within viologen-modified films indicate that the catalytic current for $H_2$ oxidation decays significantly within one day when $O_2$ is present. In contrast, the turnover stability in anaerobic experiments reaches weeks[1]. This discrepancy between experiments and theoretical predictions implies that under aerobic conditions, processes other than $O_2$-induced deactivation cause loss in catalytic current. The $O_2$ reduction by viologen moieties typically yields reactive oxygen species, in particular $H_2O_2$[19,26], which can be further converted into $H_2O$ as previously observed for viologen-modified polymer films[1]. However, $H_2O_2$ conversion to $H_2O$ is slow and is thus hypothesized to cause damage within the catalytic film. Clarification of the degradation processes under aerobic conditions is a key prerequisite for designing the redox matrices as a more robust shield against oxidative stress.

Here, we combine confocal fluorescence microscopy (CFM) and coherent anti-Stokes Raman scattering (CARS) with electrochemistry for the operando analysis of viologen-modified polymer films during catalytic turnover for $O_2$ reduction. We demonstrate that $H_2O_2$ accumulates and primarily promotes damage to the polymer backbone rather than loss of active viologen moieties or catalyst. The addition of iodide to the electrolyte enhances the aerobic half-life of viologen-modified polymer films containing an $O_2$-sensitive hydrogenase, reaching values up to 1 week under $H_2/O_2$ mixed gas feed. Our data validate the predictions from our previous model[3,25] that protection within redox films make catalysts immune to $O_2$, and demonstrate that suppression of reactive oxygen species is also necessary to avoid oxidative degradation of the catalytic films.

## Results

**Disproportionation of $H_2O_2$.** In order to decipher the causality between $H_2O_2$ generation and the loss of functionality of viologen-modified matrices in aerobic conditions, we compare the behavior of the films in the presence and absence of a catalyst for $H_2O_2$ disproportionation. While enzymes such as catalase are commonly applied catalysts for the conversion of $H_2O_2$ to water and molecular oxygen[25], their co-immobilization in the film or addition into the electrolyte do not enable sufficient stabilization of the films (see Supplementary Note 1 and Supplementary Figs. 1–3). Instead, we employ iodide (I$^-$) owing to its small size and the possibility to use it in high concentrations. Iodide added to the electrolyte can freely diffuse into the redox hydrogel film and catalyzes decomposition of $H_2O_2$ (Fig. 1b). The proposed mechanism follows Eqs. (1) and (2) in aqueous solution[27,28]:

$$H_2O_2 + I^- \rightarrow H_2O + IO^- \tag{1}$$

$$H_2O_2 + IO^- \rightarrow H_2O + I^- + O_2 \tag{2}$$

While dismutation of two molecules of $H_2O_2$ generates one molecule of $O_2$, successive cycles of the viologen-catalyzed $O_2$ reduction reaction and of the iodide-catalyzed $H_2O_2$ decomposition results in $H_2O$ as the sole end product. Iodide is regenerated in the process.

**Detection of $H_2O_2$.** We used operando confocal fluorescence to probe the $H_2O_2$ production from the viologen-modified hydrogel matrix. We exploited the intrinsic fluorescence of the viologen moieties tethered on the polymer backbone to visualize the volume of the redox matrix. $H_2O_2$ production was monitored using a fluorescent reporter (Amplite™ IR) and horseradish peroxidase added to the electrolyte. Fluorescence emission spectra of the viologen films and $H_2O_2$ fluorescent probes were recorded during $H_2O_2$ generation from electrochemical $O_2$ reduction

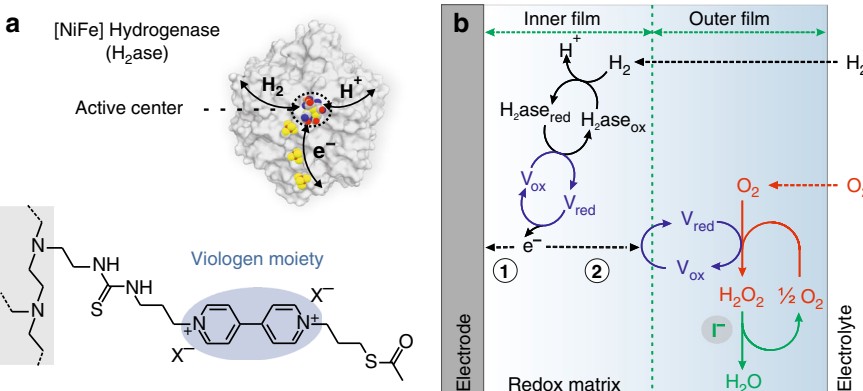

**Fig. 1 Redox matrices for the protection of $O_2$-sensitive catalysts. a** Structure of the [NiFe]-hydrogenase and the viologen-modified polymer (polyethylenimine backbone). **b** Proposed mechanism for protection from $O_2$. The hydrogenase catalyzes the reversible oxidation of $H_2$ in the inner parts of the film and generates electrons that are transferred to the electrode via the viologen ($V_{red}/V_{ox}$) moieties (pathway 1) which produces the catalytic current when an oxidative potential is applied to the electrode. In addition, the presence of $O_2$ in the electrolyte creates an oxidative driving force that diverts some of the electrons towards the electrolyte/film interface (pathway 2) where the viologen moieties act as catalysts for $O_2$ reduction, thus protecting the catalyst in the inner film domain but also producing $H_2O_2$. Iodide catalyzes the subsequent $H_2O_2$ dismutation to $H_2O$ and half a molecule of $O_2$, which is further reduced by the viologen.

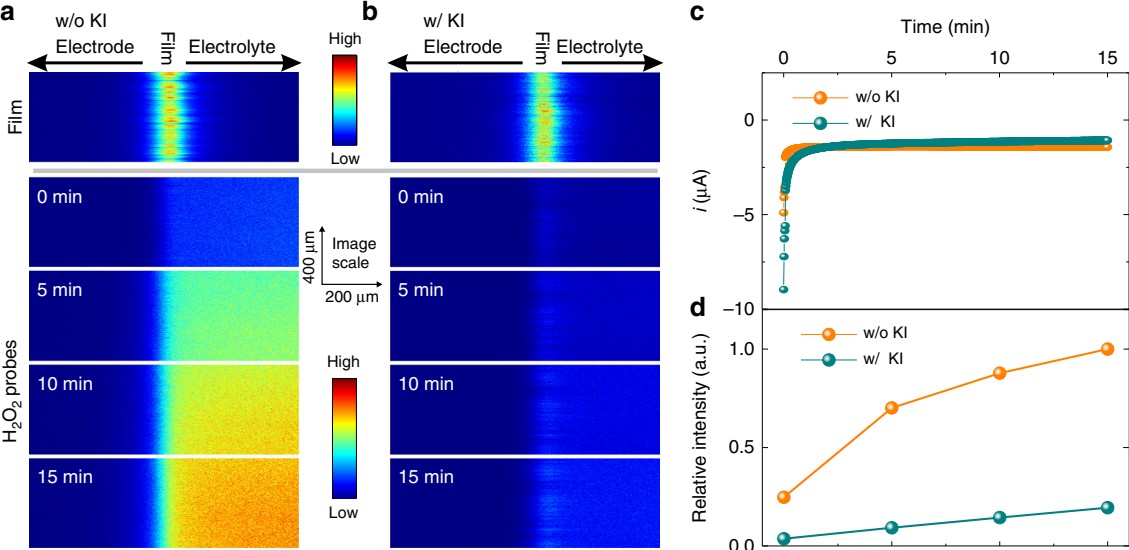

**Fig. 2 Detection of H$_2$O$_2$ using confocal fluorescence microscopy.** Representative fluorescent depth profile images of the viologen-modified films (top panel) at open circuit potential (OCP) and of H$_2$O$_2$-sensitive indicators (lower panels: 0–15 min) at a constant potential of −0.1 V vs SHE **a** in the absence and **b** in the presence of KI (0.1 M). **c** O$_2$-reduction current in absence and presence of KI (0.1 M) at constant potential of −0.1 V vs SHE. **d** The relative intensity of fluorescent H$_2$O$_2$ reporter, derived from **a** and **b**, normalized to the maximum value. An overlay plot of the depth profile of the fluorescence of the viologen and of the fluorescence of the H$_2$O$_2$-sensitive indicators is given in Supplementary Fig. 5 for better visualization of their respective positions. For all measurements, the glassy carbon electrodes (GCE, 3 mm in diameter) were coated with viologen-modified polymer with a surface coverage of 2.2 mg cm$^{-2}$. Electrolyte: 2 ml PB (0.1 M, pH 7) with 10 μL stock solution of Amplite™ Fluorimetric Hydrogen Peroxide Assay Kit. The fluorescence of the oxidized viologen-modified films was collected around its maximum emission of 590 nm upon excitation at 488 nm. The emission of the H$_2$O$_2$ fluorescent probe was determined around its maximum emission of 660 nm upon excitation at 633 nm. The scale is the same for all images. All measurements were performed under ambient air and at 300 K.

(Supplementary Fig. 4). Distinction between the matrix-based fluorescence and the fluorescence originating from the H$_2$O$_2$ reporter was made possible by measuring at different excitation and emission wavelengths, namely 488 nm excitation and 500–630 nm emission for viologen mapping, and 633 nm excitation and 640–750 nm emission for H$_2$O$_2$ monitoring. The presence of potassium iodide (KI) did not affect the absorption or emission spectra of either the H$_2$O$_2$-sensitive probe or the viologen moiety (Supplementary Fig. 4).

Confocal fluorescence images were generated as a depth profile (Fig. 2) with the intensities corresponding to the local concentrations of fluorophores[4]. At open circuit potential fluorescence was observed upon excitation of the viologen moieties (top panel in Fig. 2a, b) in both the presence and absence of iodide. Assuming the viologen moieties are evenly distributed within the film, determination of the full-width half-maximum (FWHM) from the fluorescence profiles enables an estimation of the polymer film thickness. The thickness was similar (FWHM is about 25 μm, Supplementary Fig. 5) for the electrodes in the presence and absence of iodide.

The electrodes were then held at a constant potential of −0.1 V vs SHE (standard hydrogen electrode) to induce the reduction of O$_2$, catalyzed by the viologen moieties in the film. O$_2$-reduction currents were similar for electrodes in the presence and absence of iodide, which suggests comparable H$_2$O$_2$ generation rates (Fig. 2c). Confocal fluorescence images were recorded for 15 min at 5-min intervals. In the absence of iodide, the H$_2$O$_2$-derived fluorescence rose rapidly in the electrolyte (Fig. 2a, d). In contrast, the rate of H$_2$O$_2$-mediated fluorescence rise was about sixfold lower in the presence of iodide (Fig. 2b, d). Moreover, in the absence of the polymer film, both the current for O$_2$ reduction and the fluorescence from H$_2$O$_2$ were negligible (Supplementary Fig. 6). This demonstrates that O$_2$ reduction catalyzed by the viologen-modified film generates H$_2$O$_2$, and that

H$_2$O$_2$ can be efficiently dismutated when iodide is present in the electrolyte.

**Electrochemical analysis of the film degradation.** The effect of H$_2$O$_2$ on the degradation of the viologen-modified polymer film was investigated by performing cyclic voltammetry in the presence and absence of I$^-$. Under argon atmosphere, the cyclic voltammograms (CVs) of the polymer-coated GCE showed a reversible redox wave at $E_{1/2}$ = −0.21 V vs SHE, which is attributed to the viologen moieties (Fig. 3a, b). In the presence of I$^-$, the shape of the re-oxidation peak became sharper. This is due to the higher hydrophobicity of the I$^-$ counter-anion in comparison to phosphate anions, which leads to film deswelling especially in the reduced state, thus enhancing the interactions between viologen moieties (Frumkin behavior[4]) and enabling rapid reoxidation (surface confined characteristics). Continuous cycling of CVs demonstrated that the viologen-modified polymer films were stable under anaerobic conditions in both absence and presence of I$^-$ (Supplementary Fig. 7). The addition of 5% O$_2$ to the argon supply generated a cathodic wave centered at −0.1 V vs SHE, which is more positive than the formal potential of the viologen moieties in the polymer film (Fig. 3a, b). The cathodic wave is attributed to the O$_2$ reduction reaction catalyzed by the viologen moiety in the redox hydrogel film (O$_2$ reduction directly at the bare GCE occurs at potentials 150 mV more negative[1], see Supplementary Fig. 10a). The peak shaped current response overlapping with the catalytic sigmoid curve is due to reversible reduction/oxidation of viologens that are not involved in catalysis. Electrochemical simulations demonstrate that the positive potential shift of the catalytic wave with respect to the viologen signals is in agreement with a regime of fast catalysis in thin redox-active films[29,30] (see Supplementary Note 2 and Supplementary Figs. 9 and 10b).

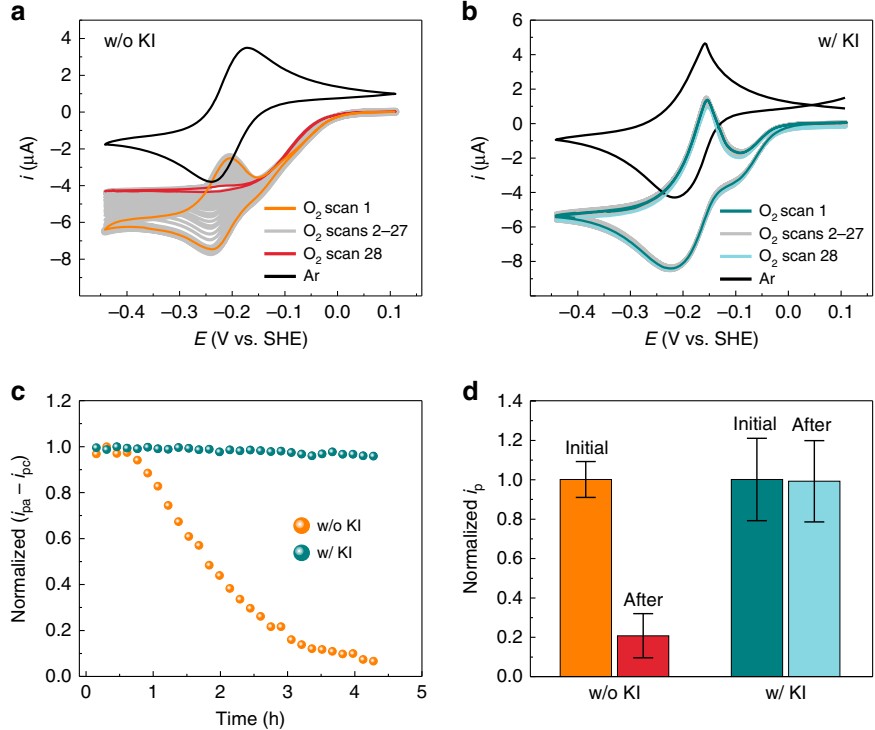

**Fig. 3 Electrochemical study of the viologen-modified film degradation.** Cyclic voltammograms (CVs, 28 cycles) of viologen-modified hydrogel films on glassy carbon electrodes (GCE) under 5% $O_2$ in argon **a** in the absence and **b** in the presence of KI (0.1 M). CVs recorded under 100% argon are shown as black curves. **c** Normalized peak current amplitudes (anodic peak current ($i_{pa}$) − cathodic peak current ($i_{pc}$)) extracted from the reversible redox signals of the viologens overlapping with the catalytic wave for $O_2$ reduction derived from the measurements shown in **a** and **b**. **d** Normalized peak current ($i_p$) before and after $O_2$ reduction derived from CVs under 100% argon (Supplementary Fig. 8). Error bars are defined as standard deviation. Average values and standard deviations were obtained from measurements at three individually prepared electrodes and then normalized by average $i_p$ values before $O_2$ reduction. All measurements were performed with GCE (3 mm in diameter) coated with viologen-modified polymer with a surface coverage of 0.3 mg cm$^{-2}$. Electrolyte: phosphate buffer (0.1 M, pH 7). Scan rate: 2 mV s$^{-1}$. Rotation rate: 2000 r.p.m. Source data are provided as a Source Data file.

Consecutive cycles in the presence of $O_2$ showed that the peak shaped current was mostly unaffected in presence of I$^-$, but it vanished almost completely within 3 h of cycling in absence of I$^-$ (Fig. 3a, c). Comparison of the peak current under argon before and after $O_2$ reduction enabled the quantification of the extent of current decrease (Fig. 3d). In the absence of I$^-$, the peak current dropped to 20% of its original value during the 4 h of measurement, while no change in peak current was detected over the same time period when I$^-$ was present in the solution during $O_2$ reduction (Fig. 3b, c). Since I$^-$ suppresses $H_2O_2$ generation, as demonstrated by the CFM investigations, the conclusion from the CV results is that loss of viologen peak current is associated with $H_2O_2$ generation.

We subsequently investigated the degradation of the viologen-modified polymer film using CARS and CFM while a potential of −0.1 V vs SHE was applied to the electrodes in the presence of $O_2$ from air (Fig. 4). CARS microscopy has intrinsic high spatial resolution and the contrast in the images results from $CH_2$ stretching vibrations and thus allows for direct monitoring of the polyethylenimine backbone of the viologen-modified polymer. Fluorescence was induced by excitation of the viologen moieties as described above. While fluorescence only reveals the oxidized viologen moieties, CARS can visualize the polymer position regardless of viologen presence or state.

Initially (at $t = 0$), CARS and fluorescence imaging showed a smooth hydrogel film without breaks both in the absence and presence of I$^-$. The co-localization of the signals from both fluorescence and CARS validated a homogeneous distribution of the viologen moieties within the polymer film. In the experiment without I$^-$, the hydrogel film underwent a distinct deformation

and started to detach from the electrode surface after 1.5 h of continuous $O_2$ reduction. Moreover, the fluorescence intensity slightly decreased, possibly because of $H_2O_2$-induced degradation of viologen moieties[31]. At later time points, the film became wrinkled (4 h, Fig. 4a) and completely detached from the electrode after 6 h (Supplementary Movies 1–4). In contrast, the hydrogel film, as well as the viologen fluorescence, remained unchanged under the same conditions in the presence of I$^-$ (Fig. 4b). This indicates that the significant loss of peak current observed in the electrochemical measurements in the absence of KI (Fig. 3) was mainly because the polymer backbone, and possibly some of the viologen moieties, degraded and detached from the electrode surface. This is consistent with the previously reported oxidation and degradation of polyethylenimine when exposed to hydrogen peroxide[32,33]. These observations highlight the need for efficient dismutation of $H_2O_2$ generated from the viologen-catalyzed $O_2$ reduction reaction in order to avoid decomposition of the redox hydrogel film.

**Turnover stability of viologen-modified films containing hydrogenase.** The impact of dismutating $H_2O_2$ with iodide on the turnover stability for enzymatic $H_2$ oxidation was investigated with electrodes coated with viologen-modified polymer and hydrogenase. These catalytic films were made sufficiently thick (polymer surface coverage: 2.26 mg cm$^{-2}$) so that the mass transport of $H_2$ limits the catalytic current as evidenced by its linear dependence on $H_2$ partial pressure[34] both in the presence and absence of I$^-$ (Supplementary Fig. 11). Under such substrate transport limitations, the theoretical models predict that the

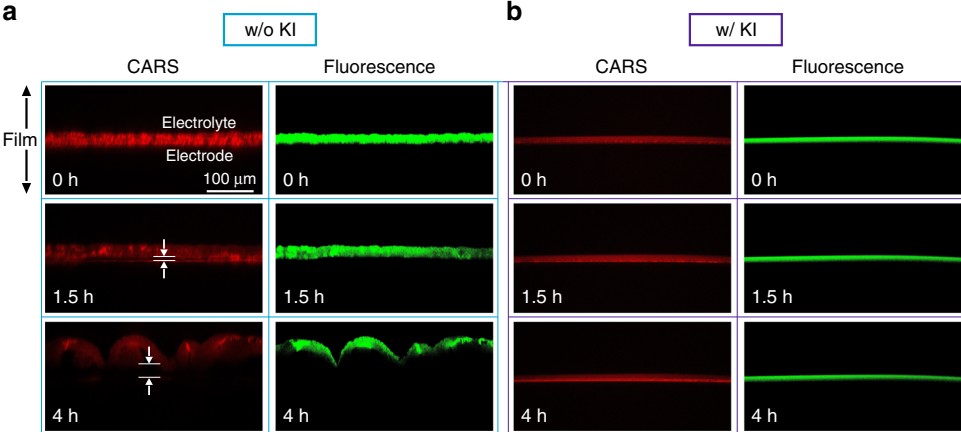

**Fig. 4 Confocal microscopy study of the viologen-modified film degradation.** Coherent anti-Stokes Raman scattering (CARS) and confocal fluorescence microscopy **a** in the absence and **b** in the presence of KI (0.1 M). A constant potential of −0.1 V vs SHE was applied to the electrodes in the presence of air to generate $H_2O_2$. CARS images show Raman intensity at 2841 cm$^{-1}$. Fluorescence was recorded between 500 and 630 nm at 488 nm excitation. The arrows in the CARS images (1.5 and 4 h) highlight the gap between the film and the electrode surface. For all experiments, the GCE (3 mm in diameter) were coated with viologen-modified polymer with a surface coverage of 2.2 mg cm$^{-2}$. Electrolyte: phosphate buffer (0.1 M, pH 7). The scale is the same for all images. These images are part of a longer time-resolved measurement available as a video file (Supplementary Movies 1–4).

hydrogenase should be protected quasi-infinitely from $O_2$[3]. Comparison of the turnover stabilities for the polymer-hydrogenase-modified electrodes under 95% $H_2$/5% $O_2$ mixed gas feed in the presence or absence of I$^-$ are given in Fig. 5a. Similar $O_2$ concentrations were recorded in both cases during the experiments (Supplementary Fig. 12). In the absence of I$^-$, only half of the initial current remained after 1.5 day (half-life: 1.4 ± 0.14 day). The loss of catalytic current is in agreement with the significant film degradation observed in the confocal microscopy studies (Fig. 4a). By contrast, in the presence of I$^-$ the loss in catalytic current was only 9% in the same time period and pro-longed measurements showed a half-life of 7.3 ± 1.9 days) (Supplementary Fig. 12b, d), which is also reached at elevated temperatures of 40 °C (Supplementary Fig. 12e half-life: 6.3 ± 1.4 days). A control experiment lacking the viologen polymer in which the enzyme is in a direct electron transfer configuration with the electrode shows that iodide concentrations up to 100 mM do not significantly impact the catalytic activity for $H_2$ oxidation under anaerobic conditions (Supplementary Fig. 14). Therefore, the strongly enhanced aerobic stability of the electro-catalytic output is attributed to the iodide-catalyzed $H_2O_2$ dis-mutation which prevents film degradation as demonstrated from the CARS and fluorescence experiments (Fig. 4b). The aerobic turnover stability in the presence of I$^-$ approaches but still does not reach that under anaerobic conditions (6 weeks)[1]. This dif-ference can be explained by the presence of residual $H_2O_2$ as observed by operando CFM (Fig. 2d). Altogether, these results suggest that $H_2O_2$ is the main reason for the decrease of the catalytic current from polymer-hydrogenase electrodes exposed to $O_2$.

The applicability of the iodide containing electrolyte in $H_2/O_2$ biofuel cells was tested by coupling the hydrogenase modified anode to an $O_2$ reducing cathode modified with the enzyme bilirubin oxidase in a two-compartment cell. The anode compart-ment was filled with phosphate buffer containing iodide and the cathode compartment with phosphate buffer only to avoid iodide oxidation. The fuel cell performance tests yielded current densities of 117 ± 30 µA cm$^{-2}$, power density of 5.8 ± 1.2 µW cm$^{-2}$, and an open circuit voltage of 1.0 ± 0.05 V (Fig. 5b). These performances are comparable with the ones obtained from fuel cells that do not contain iodide[1,2] and thus demonstrate the absence of detrimental effects of its use as electrolyte in the anodic compartment.

## Discussion

The discrepancy between model predictions and experimental observations for the half-life of $O_2$-sensitive catalysts embedded in viologen-modified films used as protection matrices has been revealed through a combination of operando CFM and CARS coupled with electrochemistry. We demonstrate that the use of iodide in the electrolyte leads to suppression of $H_2O_2$, to enhanced stability of the viologen polymer film when catalyzing $O_2$ reduction, and to extended half-life of the hydrogenase when catalysing $H_2$ oxidation (up to 1 week) under aerobic conditions. These results indicate that the loss in catalytic current for $H_2$ oxidation is not due to the direct deactivation of the hydrogenase by $O_2$, but rather due to the degradation of the redox matrix caused by $H_2O_2$ generated from the $O_2$ reduction reaction when catalyzed by the viologen-modified polymer. This is in agreement with the theoretical model that shows that the polymer film efficiently protects the hydrogenase from $O_2$ by reducing this deactivating molecule at the film–electrolyte interface before it reaches the $O_2$-sensitive catalyst within the film[3]. However, in order to achieve full protection, complete reduction of $O_2$ to water is necessary to bypass additional deactivation pathways involving reactive oxygen species that degrade the redox film.

In this context, the use of iodide is particularly advantageous since the dismutation of $H_2O_2$ leads to water and half a molecule of $O_2$ that is further reduced by the viologen, and thus eventually leads to water as the sole final product. The iodide being used as a catalyst in this reaction is not used up and therefore represents a practical solution for long-term operation of energy converting systems in which $H_2O_2$ production is often an undesired side reaction of the system. This approach for $H_2O_2$ removal being exclusively based on tuning the composition of the electrolyte, rather than tuning the catalytic system itself, is directly applicable for enhancing the stability of other low potential electrochemical or photochemical systems suffering from degradation induced by $H_2O_2$ generation.

## Methods

**Chemicals, polymer synthesis and protein purification.** Amplite™ Fluorimetric Hydrogen Peroxide Assay Kit (Near Infrared Fluorescence) (AAT Bioquest), potassium iodide, tris(hydroxymethyl)aminomethane, potassium dihydrogen, and di-potassium hydrogen phosphate trihydrate were purchased from VWR Chemi-cals. All chemicals were of analytical grade and directly used as received without further purification. The viologen-modified polymer was synthesized as described

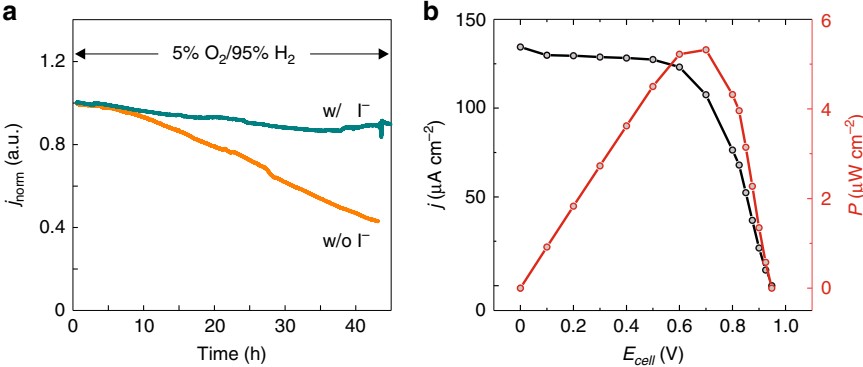

**Fig. 5 Turnover stability of viologen-modified films containing hydrogenase. a** The catalytic current densities for $H_2$ oxidation in the presence of 5% $O_2$ are normalized ($j_{norm}$) by their maximum values. The measurements are performed in phosphate buffer (0.1 M, pH 7) in the absence or in the presence of KI (0.1 M). Replicates of measurements over extended time periods under the same conditions are given in Supplementary Fig. 12. Measurement conditions: The potential was held at 0.21 V vs SHE. The rotation rate was 2000 r.p.m. **b** Current density (black trace) and power density (red trace) vs cell voltage from fuel cell measurements conducted in a two-compartment cell with a glass frit as a separator. The anode modified with a film of the viologen polymer containing the hydrogenase was placed in the compartment containing phosphate buffer (0.1 M, pH 7) and KI (0.1 M) purged with 100% $H_2$. The cathode modified with bilirubin oxidase for $O_2$ reduction was placed in the compartment containing phosphate buffer (0.1 M, pH 7) purged with 100% $O_2$. Replicates of the fuel cell measurements under the same conditions are given in Supplementary Fig. 13. The polymer surface coverage for all hydrogenase modified electrodes is 2.26 mg cm$^{-2}$. All measurements were performed at 298 K. Source data are provided as a Source Data file.

previously[1] and dissolved (40 mg ml$^{-1}$) in distilled water for drop-casting. DvMF [NiFe] hydrogenase was purified as described previously[35,36].

**Film formation on glassy carbon electrode surface.** For hydrogenase-polymer electrodes, an aqueous suspension of the viologen-modified polymer (4 μL, 40 mg ml$^{-1}$) and [NiFe] hydrogenase from *Desulfovibrio vulgaris* Miyazaki F (3 μL, 200 μM in 0.01 M MES buffer, pH 6.8) were mixed and applied to the GCE (3 mm diameter). For CFM, CV, and fuel cells studies, the polymer was drop-cast onto the GCE with surface coverages indicated in the captions of the figures. In all cases, Tris buffer was added to the solution droplet (1 μL, 0.1 M, pH 9) immediately after drop-casting to accelerate the polymer cross-linking via disulfide bond formation. Electrodes were stored at 4 °C in a closed vessel overnight. After 12 h the solution became turbid and was then left to dry in air for another hour before the electrochemical measurements.

**Fluorescence characterization.** Fluorescence measurements were performed on a Leica Microsystems TCS SP8 CARS laser scanning microscope in a fully confocal optical setup. An argon gas laser excited the samples at 488 nm and an He–Ne laser was used for excitations at 633 nm, through an HC PL Fluotar ×10/0.3 dry or Fluotar VISIR ×25/0.95 water objective. The emission was detected in EPI direction with photomultiplier tubes (PMTs) from 500 to 630 nm, and from 640 to 750 nm, respectively. During fluorescence measurements the temperature in the sample compartment was stabilized at 300 K. The stock solutions of Amplite™ fluorimetric hydrogen peroxide assay kit were prepared according to the manufacturer's protocol.

**CARS microscopy.** CARS measurements were performed on a Leica Microsystems TCS SP8 CARS laser scanning microscope. A picoEmerald laser system from APE Berlin generated two synchronized pulsed laser beams at 817 and 1064 nm, respectively, which were temporally and spatially overlapped at the measurement position on the sample. The resulting CARS signal at 663 nm was detected via a non-descanned PMT. This corresponds to a Raman intensity at 2841 cm$^{-1}$. The detection window of the PMT ranges from 560 to 750 nm. During CARS measurements the temperature in the sample compartment was stabilized at 300 K. The CARS lasers and signal detection were focused and collected via the same objectives as for the fluorescence measurements. The measurements followed each other in close succession (very few seconds).

**Electrochemistry.** All electrochemical measurements were carried out with BA Metrohm Autolab as well as a rotating-disc electrode setup. Gas mixtures of $H_2$, Ar, and $O_2$ were controlled with mass-flow controllers. A Pt wire was used as counter electrode. The reference electrode was Ag/AgCl/3 M KCl. Potentials are converted to the SHE using the correction $E_{SHE} = E_{Ag/AgCl} + 210$ mV. All electrochemical measurements were performed at 298 K. During the measurements, the $O_2$ concentration was monitored with FireStingGO2.

The $H_2$/$O_2$ biofuel cell was assembled from a hydrogenase-polymer modified anode and a bilirubin oxidase modified cathode (see ref. [37] for cathode preparation) in a two-compartment cell with a porous glass frit as a separator. The anode for $H_2$ oxidation was placed in the compartment containing phosphate buffer (0.1 M, pH

7) and KI (0.1 M) purged with 100% $H_2$. The cathode for $O_2$ reduction was placed in the compartment containing phosphate buffer (0.1 M, pH 7) purged with 100% $O_2$. The fuel cell was operated under anode-limiting conditions by using an oversized bilirubin oxidase modified cathode[1,2]. Power curves were calculated from the steady-state currents obtained from stepped potential chronoamperometric experiments.

## Data availability
All data that support the findings of this study are available from the corresponding author upon request. The source data underlying Fig. 3, Fig. 5, Supplementary Figs. 1–3, Supplementary Figs. 7, 8, Supplementary Fig. 12 and Supplementary Fig. 14 are provided as a Source Data file. The code for the model described in Supplementary Note 2 is provided as a Source Data file.

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

## Acknowledgements

The authors thank Dr. Tobias Vöpel for discussions about the H2O2 reporter, Stefanie Stapf for polymer synthesis, Steffen Hardt for help with the fuel cell experiments, and Nina Breuer for the preparation of the DvMF [NiFe] hydrogenase. This work was supported by RESOLV, funded by the Deutsche Forschungsgemeinschaft (DFG, German Research Foundation) under Germany´s Excellence Strategy—EXC-2033—Projektnummer 390677874, by the ERC starting grant 715900 and by the ANR-DFG project SHIELDS (PL 746/2-1). H.L. is grateful for the support by the China Scholarship Council (CSC). U.M. and E.F. acknowledge funding by the "Ministerium für Kultur und Wissenschaft des Landes Nordrhein-Westfalen", the "Regierender Bürgermeister von Berlin-inkl. Wissenschaft und Forschung", and the "Bundesministerium für Bildung und Forschung", also in the form of the Leibniz-Research-Cluster (grant number: 031A360E).

## Author contributions

N.P., H.L., and E.F. conceived the research. H.L performed all electrochemical experiments involving mediated electron transfer and contributed to the confocal fluorescence experiments. A.A.O contributed with direct electron transfer electrodes. D.B. performed the electrochemical modeling and simulations. E.F. and U.M. conceived and performed the spectroscopic experiments. W.L. contributed the hydrogenase. All authors contributed to writing the manuscript.

## Competing interests

The authors declare no competing interests.
