## [Peer Review File · Nature Communications]

Reviewers' comments:

Reviewer #1 (Remarks to the Author):

In this MS, by combining in operando confocal fluorescence microscopy and coherent anti-Stokes Raman scattering coupled with electrochemistry, the authors revealed the deactivation process occurring for hydrogenase immobilized within viologen based redox polymer. It is attributed to the production of hydrogen peroxide which induces the decomposition of the polymer backbone and to some extent to the denaturation of the viologen moieties rather to the enzyme deactivation. The addition of iodide, which decomposes hydrogen peroxide, supports this hypothesis. This is a very nice paper, well written. This work may seem incremental in comparison with the previous published papers by the same author. But, a set of new analytical techniques have been used to identify the reason for the electrode deactivation and may be extended to other electrochemical systems. This would justify the publication in Nature Communications after a couple of corrections

1- The authors are employing iodide because it is smaller than catalase arguing that I⁻ could diffuse into the redox hydrogel to scavenge hydrogen peroxide. However, HRP is used for the monitoring of hydrogen peroxide. Therefore, catalase could have also been used in the electrolyte with the same effect than iodide. The choice of iodide over catalase is not clear for this referee and this point would need to be clarified by the authors

2- Figure 4. Are we seeing the redox polymer leaching from the electrode surface or is it possible to envision that the polymer backbone denaturates, losing the CH₂ vibration and therefore the signal, but still remain attached to the electrode surface.

3- Figure S3b: The charge seen in this CV is much bigger than in Fig S3a and does not seem to be in agreement with the CV seen in FigS3d and Figure 3b.

4- Error bars are missing and the relative standard deviation of at least three experiments should be calculated.

5- Is there any effect of iodide on the hydrogenase?

6- Caption Figure 5, experiment in presence of KI. The color looks green and not blue

Reviewer #2 (Remarks to the Author):

Kudos to the authors on this excellent work. The combination of electrochemical and spectroscopic techniques to identify and address performance degradation in redox films will likely be of interest to readers of Nature Communications and will likely influence thinking in related fields of research. The comments and questions of this reviewer are outlined below:

1) In electrochemical experiments involving the viologen coated electrodes without hydrogenase, can the authors integrate the return redox features measured under argon and under O₂ to determine the percentage of redox active viologens that are participating in catalysis? Further, can the authors comment (based on the total viologen polymer loadings) on the fraction of viologens that are redox active?

2) In electrochemical experiments involving electrodes containing both hydrogenase and the viologen polymer, are the authors able to quantify the concentrations of H₂O₂ being produced (both with and without the presence of I⁻)?

3) In the confocal fluorescence microscopy results shown in Figure 2, can the authors comment on why there appears to be relatively low fluorescence intensity when probing for H₂O₂ at the polymer film/electrolyte interface as compared to the bulk solution? Also, for the scale indicating 'high' and 'low' that is shown in Figure 2, are the authors able to include quantitative information regarding the related concentrations of H₂O₂?

4) In figure 1, the authors propose two pathways for electron flow. In one pathway, electrons generated by oxidation of hydrogen are transferred to the electrode via viologen moieties. In the other pathway, electrons used to generate reduced viologens go on to reduce O₂ to H₂O₂. It may be helpful to readers if the authors are able to add further commentary regarding the driving forces for preferring one pathway over the other.

Related to this point, in the experiments involving electrodes functionalized with viologen polymers and hydrogenase enzymes that are polarized at an applied potential (+0.21 V vs SHE) that is ~400 mV positive of the viologen midpoint potential (-0.21 V vs SHE), can the authors comment on the fraction of viologen species present in their oxidized form (which would not be able to reduce O₂)?

5) Out of curiosity (and not necessarily a requirement for any publication), have the authors performed control experiments using hydrogenase modified electrodes immobilized in a non-viologen containing polymer (for example polyethylenimine) and if so, how susceptible are the immobilized hydrogenases to O₂ under such conditions?

Reviewer #3 (Remarks to the Author):

This manuscript by Professor Plumere et al investigates the deactivation of the O₂-sensitive hydrogenase in the redox active polymer film and finds out that H₂O₂ generation from the incomplete O₂ reduction induces the decomposition of the redox matrix and leads to the loss of the hydrogenase catalyst. The addition of iodide to the electrolyte allows the removal of H₂O₂ and therefore enhances the lifetime of the H₂/O₂ biofuel cell in presence of O₂ greatly. In general, this study contains useful and novel information to construct a stable hydrogenase-based bioelectrode immune to O₂ attack and can shed light on the deactivation mechanism of such bioelectrode under O₂ gas feed. However, I feel that the impact and contribution to the field is not as significant as the previous publications from the same authors (Nat. Chem. 2014,6:822, Angew. Chem. Int. Ed. 2015,54,12329, Nat. Comm. 2018,9:864, Nat. Comm. 2018,9:4715). Moreover, more control experiments are needed and some confusions should be clarified.

(1) The control electrode containing a polymer film without viologen moiety may be constructed and evaluated bioelectrochemically.

(2) H₂O₂ can be added separately to the electrolyte and its damage to the polymer film electrode should be characterized. Then catalase can be added to see whether the damage can be mitigated.

(3) The content in line 106–112 reads confusing. What is the difference between Fig S4 and Fig S3c and d? Why they look like different at the same condition? Where is the cathodic wave centered at -0.1 V vs SHE? I don't see the more positive shift from these figures.

(4) Can KI be added onto the electrode film directly? Is there any way to minimize the loading of KI?

(5) More data should be provided to evaluate the performance of the H₂/O₂ biofuel cell when KI was used to protect the viologen polymer film. How about the influence of the high operating temperature?

We thank the three reviewers for their constructive comments that helped us to significantly improve the manuscript. In the following are our point to point answers (in blue) to the reviewer comments (in black) and our changes to the manuscript (highlighted in yellow).

Reviewers' comments:

Reviewer #1 (Remarks to the Author):

In this MS, by combining in operando confocal fluorescence microscopy and coherent anti-Stokes Raman scattering coupled with electrochemistry, the authors revealed the deactivation process occurring for hydrogenase immobilized within viologen based redox polymer. It is attributed to the production of hydrogen peroxide which induces the decomposition of the polymer backbone and to some extent to the denaturation of the viologen moieties rather to the enzyme deactivation. The addition of iodide, which decomposes hydrogen peroxide, supports this hypothesis. This is a very nice paper, well written. This work may seem incremental in comparison with the previous published papers by the same author. But, a set of new analytical techniques have been used to identify the reason for the electrode deactivation and may be extended to other electrochemical systems. This would justify the publication in Nature Communications after a couple of corrections

1- The authors are employing iodide because it is smaller than catalase arguing that I⁻ could diffuse into the redox hydrogel to scavenge hydrogen peroxide. However, HRP is used for the monitoring of hydrogen peroxide. Therefore, catalase could have also been used in the electrolyte with the same effect than iodide. The choice of iodide over catalase is not clear for this referee and this point would need to be clarified by the authors

We now demonstrate experimentally that catalase is less efficient than iodide for protection of the viologen modified film from the detrimental effect of peroxides. We attribute the superior performance of iodide to the possibility to use it at high concentration owing to high solubility and its ability to penetrate into the film owing to its small size compared to the large catalase proteins.

We added the following in the main text as well as a new section in the Supplementary Information showing the film stability in presence of catalase:

“While enzymes such as catalase are commonly applied catalysts for the conversion of H₂O₂ to water and molecular oxygen,²⁵ their co-immobilization in the film or addition into the electrolyte do not enable sufficient stabilization of the films (see SI Section 1).“

2- Figure 4. Are we seeing the redox polymer leaching from the electrode surface or is it possible to envision that the polymer backbone denaturates, losing the CH₂ vibration and therefore the signal, but still remain attached to the electrode surface.

The C-H stretching vibration was chosen due to its strong signal and therefore nice imaging capabilities. If the polymer backbone partly denaturates, we would still detect a signal, even of a barely intact film. Therefore, the loss of signal in the CARS images in Fig. 4 w/o KI is a strong evidence of loss of film at this position. The change in CARS (as well as fluorescence) images reveal that the film completely loses its cohesion and detaches from the electrode which is consistent with degradation of the polymer backbone.

Moreover we added in the main text the following comment:

“This is consistent with the previously reported oxidation and degradation of polyethylenimine when exposed to hydrogen peroxide”

29. Haas, H. C., Schuler, N. W. & Macdonald, R. L. Oxidized polyethylenimine. *J. Polym. Sci., Part A-1: Polym. Chem.* **10**, 3143–3158; 10.1002/pol.1972.170101102 (1972).

30. Englert, C. *et al.* Enhancing the Biocompatibility and Biodegradability of Linear Poly(ethylene imine) through Controlled Oxidation. *Macromolecules* **48**, 7420–7427; 10.1021/acs.macromol.5b01940 (2015).

3- Figure S3b: The charge seen in this CV is much bigger than in Fig S3a and does not seem to be in agreement with the CV seen in Fig. S3d and Figure 3b.

We use relatively thick films (about 25 μm according to confocal microscopy images in Fig. 4) to ensure sufficient protection from O₂ in the case of the hydrogenase experiments and more generally because our attempts to make thinner films with this viologen modified polymer typically lead to less homogeneous thicknesses (see ref 4). In the case of this thick film, even at 2 mV/s, the scan rate is not slow enough to fully reduce and oxidize the films at the end of each scan (note that the outer layer of the films are reduced when hydrogenase is present due to H₂ oxidation). This leads to current response with diffusion characteristics (apparent diffusion due to electron hopping within the film). Under such conditions the current depends on the apparent diffusion coefficient of the electron and on the concentration of the viologen (Randles Sevcic equation).

In comparison to phosphate, iodide is a relatively hydrophobic counterions for the positively charged viologen. It partially desolvates the films which shrinks compared to the same film in phosphate buffer only. This is in agreement with the film thickness observed in Fig. 4 by confocal microscopy for films with and without KI. The film shrinking induces higher viologen concentrations and shorter distances between the viologen which increase the electron hopping

processes. This explains why the peak currents (and charge) are higher (200% more) for the films upon addition of KI (Fig. S3a vs Fig. S3b).

We summarize this as follow in the caption of Fig. S7

“The peak currents in presence of KI are typically higher compared to the measurements carried out in pure PB most likely because iodide as a hydrophobic counter-ion partially desolvates the film which increases the concentration of the viologen moieties and electron hopping rates.”

The difference between S3b and S3d (current is 60 % higher in S3d) are attributed to batch to batch variation in polymer preparations and in film formations.

This also explains the difference with Fig. 3b although comparison is less straightforward here since some of the viologen are involved in O₂ reduction and thus not involved in reversible oxidation/reduction.

We have repeated the measurements corresponding to Fig. 3 and Fig. S3. The average values and standard deviation from 3 replicates are added in the Fig. 3 or in the Fig. caption (Fig. S7).

4- Error bars are missing and the relative standard deviation of at least three experiments should be calculated.

We have added error bars and/or standard deviations resulting from at least 3 replicates in the following figures:

- Fig. 3: Electrochemical analysis of the effect of iodide on viologen-modified film degradation.
- Fig. 5: Turnover stability of viologen-modified films containing hydrogenase. (half life standard deviation added in the main text)
- Fig. S1: Effect of catalase in solution on film degradation during viologen catalyzed O₂ reduction.
- Fig. S2: Effect of catalase in the film on film degradation during viologen catalyzed O₂ reduction.
- Fig. S3: The effect of H₂O₂ added to the electrolyte on viologen polymer degradation.
- Fig. S7: CVs of viologen-modified polymer films on GC electrodes in anaerobic conditions. (Fig. S3a, b in first version, standard deviation added in caption)
- Fig. S8: CVs of viologen-modified polymer films on GC electrodes in aerobic conditions. (Fig. S3c, d in first version, standard deviation added in caption)
- Fig. S14: Effect of iodide on the catalytic activity of enzyme for H₂ oxidation.

5- Is there any effect of iodide on the hydrogenase?

In order to study the effect of iodide on the hydrogenase, we examined the enzyme in a direct electron transfer (DET) configuration. An electrode was covalently modified with the hydrogenase and catalytic H₂ oxidation activity was measured with and without the addition of KI to the buffer. KI titration data (Fig. S14) show that the catalytic activity stayed almost unchanged upon the addition of KI concentrations from 20 - 100 mM.

We discuss this additional data in the main text as follow:

“A control experiment lacking the viologen polymer, in which the enzyme is in a direct electron transfer configuration with the electrode, shows that iodide concentrations up to 100 mM do not significantly impact the catalytic activity for H₂ oxidation under anaerobic conditions (Fig. S14). Therefore, the strongly enhanced aerobic stability of the electrocatalytic output is attributed to the iodide catalysed H₂O₂ dismutation ...”

Fig. S14: Effect of iodide on the catalytic activity of enzyme for H₂ oxidation. (a) Titration of the buffer solution with KI of 20, 40, 60, 80 and 100 mM. Three individual CVs were taken for each concentration; the current was recorded at -150mV (vs Ag/AgCl 3M KCl) and plotted against KI concentration. (b) Cyclic voltammetry experiments of pyrolytic graphite electrodes without enzyme (solid line) and covalently modified with hydrogenase in DET configuration (dotted lines) with the addition of 0.1 M KI (red traces) and without KI (black traces). Scan rate: 10 mV s⁻¹. Electrolyte: phosphate buffer (0.1 M, pH 7), 100% H₂.

6- Caption Figure 5, experiment in presence of KI. The color looks green and not blue

We agree that the color description is subjective. We now omit the color reference.

Reviewer #2 (Remarks to the Author):

Kudos to the authors on this excellent work. The combination of electrochemical and spectroscopic techniques to identify and address performance degradation in redox films will likely be of interest to readers of Nature Communications and will likely influence thinking in related fields of research. The comments and questions of this reviewer are outlined below:

1) In electrochemical experiments involving the viologen coated electrodes without hydrogenase, can the authors integrate the return redox features measured under argon and under O₂ to determine the percentage of redox active viologens that are participating in catalysis? Further, can the authors comment (based on the total viologen polymer loadings) on the fraction of viologens that are redox active?

This would indeed be interesting. However, we use relatively thick films (about 25 μm according to confocal microscopy images in Fig. 4) to ensure sufficient protection from O₂ in the case of the hydrogenase experiments and more generally because our attempts to make thinner films with this viologen modified polymer typically lead to less homogeneous thicknesses (see ref 4). In the case of this thick film, even at 2 mV/s, the scan rate is not slow enough to fully reduce and oxidize the films at the end of each scan. For this reason it is not possible to determine the percentage of the redox active viologens that participate in catalysis from the CV experiments.

2) In electrochemical experiments involving electrodes containing both hydrogenase and the viologen polymer, are the authors able to quantify the concentrations of H₂O₂ being produced (both with and without the presence of I⁻)?

Unfortunately, our fluorescence measurements do not allow an absolute quantification, just a relative one. Even though the reporter system (Amplite Fluorimetric Hydrogen Peroxide Kit) is originally intended for quantification when used in homogeneous systems, the procedures for such quantification are not applicable to our spatially and time-resolved measurements of surface confined systems.

3) In the confocal fluorescence microscopy results shown in Figure 2, can the authors comment on why there appears to be relatively low fluorescence intensity when probing for H₂O₂ at the polymer film/electrolyte interface as compared to the bulk solution? Also, for the scale indicating 'high' and 'low' that is shown in Figure 2, are the authors able to include quantitative information regarding the related concentrations of H₂O₂?

The fluorescence is actually not lower within the film. We now add in the supplementary information an overlay that outlines the film fluorescence profile to visualize its position with respect to the fluorescence profile of the probe (Fig. S5). In respect to the position of their rise they are actually very similar.

We also included the following sentence in the caption of Fig. 2:

“An overlay plot of the fluorescence intensity depth profile for viologen and for the H₂O₂ probe is given in Fig. S5 for better visualization of their respective positions.”

Fig. S5: Confocal fluorescence microscopy of viologen film and H₂O₂ detection for indicated time intervals. a) w/o KI and b) w/ KI in the electrolyte. The fluorescence of the film was derived from the viologen moieties in the oxidized state (at open circuit potential in air) upon excitation at 488 nm and detected around its maximum emission of 590 nm. Starting from 0 min, a constant potential at -0.1 V_{SHE} was applied to the electrodes. The emission of the H₂O₂ fluorescent probe was determined around its maximum emission of 660 nm upon excitation at 633 nm. The measurements were performed with GCE (3 mm in diameter) in phosphate buffer (0.1 M, pH 7) under ambient air at 300 K.

Information regarding the quantification of H₂O₂ can unfortunately not be added (see answer to point 2).

4) In figure 1, the authors propose two pathways for electron flow. In one pathway, electrons generated by oxidation of hydrogen are transferred to the electrode via viologen moieties. In the other pathway, electrons used to generate reduced viologens go on to reduce O₂ to H₂O₂. It may be helpful to readers if the authors are able to add further commentary regarding the driving forces for preferring one pathway over the other.

This is an excellent suggestion, we specified the caption of Fig. 1 as follows:

“...generates electrons that are transferred to the electrode via the viologen (V_{red}/V_{ox}) moieties (pathway 1) which produces the catalytic current when an oxidative potential is applied to the electrode. In addition, the presence of O₂ in the electrolyte creates an oxidative driving force that diverts some of the electrons towards the electrolyte/film interface (pathway 2) ...”

Related to this point, in the experiments involving electrodes functionalized with viologen polymers and hydrogenase enzymes that are polarized at an applied potential (+0.21 V vs SHE) that is ~400 mV positive of the viologen midpoint potential (-0.21 V vs SHE), can the authors comment on the fraction of viologen species present in their oxidized form (which would not be able to reduce O₂)?

The fraction of viologen moieties present in the oxidized form during catalytic turnover, depends on three parameters, namely the H₂ concentration, the O₂ concentration and the applied potential. They define the rate of viologen reduction by the hydrogenase and the rate of viologen oxidation by oxygen and by the electrode. The balance of these two reaction rates (which are defined by electron transfer, catalytic and mass transport properties within the film) defines redox gradient and reaction layers within the film. The film is fully reduced in the middle part, fully oxidized at the electrode interface and at the electrolyte interface (see JACS 2019, 141, 16734). The progression of O₂ into the film means that the fraction of the oxidized viologen is a function of time (and of the film thickness). Based on the thickness of the reaction layers (about 3 μm) reported in Chem. Sci. (2018, 9, 7596), and based on the penetration depth of O₂ after several hours of operation (also about 2-3 μm), the fraction of reduced viologen in a film of about 25 μm, would be around 75%.

5) Out of curiosity (and not necessarily a requirement for any publication), have the authors performed control experiments using hydrogenase modified electrodes immobilized in a non-viologen containing polymer (for example polyethylenimine) and if so, how susceptible are the immobilized hydrogenases to O₂ under such conditions?

This is a very interesting suggestion to further validate the protection requirement proposed previously. We have now performed two sets of experiments based on hydrogenase immobilized in a polyethylenimine film. In the first case we added 5 mM viologen in solution which led to a reasonable catalytic current for H₂ oxidation (see figure below). Upon addition of O₂ the current decrease very rapidly which reveals that the freely diffusing nature of the viologen does not enable efficient protection as demonstrated for viologen modified polymers. This may be due to fast transportation of the reduced viologen away from the hydrogenase due to forced convection which dilutes the reducing force and thus enable access of O₂ to the hydrogenase. It may also simply be explained by the fact that the high concentration of viologen in the viologen modified films (100 mM) can not easily be applied when using the viologen in diffusion. However, a theoretical model for freely diffusing viologens for protection is not in place yet (in contrast to viologen modified films) which prevents a quantitative demonstration of this hypothesis.

We also tried to repeat the same experiment with hydrogenase in PEI and in absence of viologen in solution. However, no catalytic current was observed, which highlights the need of the

viologen not only for protection but also for shuttling the electrons between the hydrogenase and the electrode.

Overall, these experiments confirm our general understanding of protection matrices for hydrogenase but we feel that this information is somehow out of the scope of the present manuscript (H_2O_2 induced deactivation). Hence we would prefer to not use this experiment in the revision.

Electrochemical analysis of the effect of non-viologen containing polymer on H_2ase protection. The response of hydrogenase immobilized in PEI polymer film to O_2 in electrolyte (a) w/ and (b) w/o 5 mM methyl viologen. All measurements were performed with GCE (3 mm in diameter) coated with polymer with a surface coverage of 2.26 mg cm^{-2} . Measurement conditions: 5% O_2 in H_2 , electrolyte: Phosphate buffer (0.1 M, pH 7), rotation rate: 2000 rpm. The potential was held at 0.21 V vs SHE. All measurements were performed at 298 K.

Reviewer #3 (Remarks to the Author):

This manuscript by Professor Plumere et al investigates the deactivation of the O_2 -sensitive hydrogenase in the redox active polymer film and finds out that H_2O_2 generation from the incomplete O_2 reduction induces the decomposition of the redox matrix and leads to the loss of the hydrogenase catalyst. The addition of iodide to the electrolyte allows the removal of H_2O_2 and therefore enhances the lifetime of the H_2/O_2 biofuel cell in presence of O_2 greatly. In general, this study contains useful and novel information to construct a stable hydrogenase-based bioelectrode immune to O_2 attack and can shed light on the deactivation mechanism of such bioelectrode under O_2 gas feed. However, I feel that the impact and contribution to the field is

not as significant as the previous publications from the same authors (Nat. Chem. 2014,6:822, Angew. Chem. Int. Ed. 2015,54,12329, Nat. Comm. 2018,9:864, Nat. Comm. 2018,9:4715). Moreover, more control experiments are needed and some confusions should be clarified.

We thank the reviewer for these very useful comments on both the didactic and technical aspect of our manuscript.

We now clarify the impact with respect to the state-of-the-art:

Until now the research direction focussed on the proof of concept (Nat. Chem. 2014,6:822), elucidation of the protection mechanism (Angew. Chem. Int. Ed. 2015,54,12329, JACS, 2015, 137, 5494, JACS 2019,141 (42), 16734) and initial attempts toward applications (Nat. Comm. 2018,9:864, Nat. Comm. 2018,9:4715). These papers were in part published by us and also by others.

While these previous reports predicted the possibility for long term protection of O₂ sensitive catalysts from O₂, the experimental half life only reached a time scale of hours, which raises the question of the validity of the overall concept. The present manuscript explains for the first time the discrepancy between theoretical predictions and experimental results and achieves stabilization of the catalysts for more than a week under constant O₂ exposure. These breakthroughs validate the hypotheses proposed in the past 5 years and demonstrate the practicability of the system with a solution that is applicable generally in electrochemical systems sensitive to O₂ and reactive oxygen species.

(1) The control electrode containing a polymer film without viologen moiety may be constructed and evaluated bioelectrochemically.

This is indeed an important comment. We performed this experiment and the corresponding data and discussion are presented in the answer to point 5 of reviewer 2.

(2) H₂O₂ can be added separately to the electrolyte and its damage to the polymer film electrode should be characterized. Then catalase can be added to see whether the damage can be mitigated.

Addition of 10 mM of H₂O₂ to the electrolyte and cycling reduction/oxidation of the viologen polymer lead to viologen degradation of almost 100% within 28 cycles (Fig. S3). The experiment confirms the H₂O₂ degrades the viologen polymer. The addition of catalase almost completely avoids degradation when H₂O₂ is present in solution. In contrast when H₂O₂ is produced within the film it leads to viologen degradation despite the presence of catalase in solution (see Fig. S1).

(3) The content in line 106–112 reads confusing. What is the difference between Fig S4 and Fig S3c and d? Why they look like different at the same condition? Where is the cathodic wave centered at -0.1 V vs SHE? I don't see the more positive shift from these figures.

Fig. S4 (now Fig. S8) is in presence of O_2 while Fig. S3 (now Fig. S7) is under anaerobic conditions.

In homogeneous catalytic reactions O_2 reduction coupled to the electrochemical reduction of the viologen would normally show a catalytic wave centered at the non-catalytic response. In our case the catalysis takes place in the film and is fast with respect to mass transport of O_2 . The CV shows a catalytic wave overlapping with a non-catalytic signal which are shifted from each other. This has not been described previously. We now add a theoretical model to explain the current response.

We now also indicate the position of the cathodic wave with dashed lines in Fig. S10a.

We add the following to the main text:

“Electrochemical simulations demonstrate that the positive potential shift of the catalytic wave with respect to the overlapping non-catalytic signals is in agreement with a regime of fast catalysis in thin redox films (See Supplementary Section 2 Fig. S9, Fig. S10b).”

We add the following section in the SI:

Modeling and Simulation Description

Fig. S9: Reaction schematic for simulation the current response from catalysis in thin films.

Within the film ($l = 10 \mu\text{m}$) which is drop-casted onto an electrode with surface area A , the redox active Viologen moieties (concentrations C_O and C_R , total concentration O_{Tot}) are confined to the film and are assumed to undergo rapid heterogeneous electron transfer kinetics at the electrode interface. At the beginning of the experiment, all viologen moieties are assumed to be in the oxidized state. As the potential is linearly cycled between the initial and final potential values, R is produced at the electrode surface. As R diffuses (apparent diffusion with a diffusion coefficient D_R) through the film by means of electron hopping, it irreversibly reacts (second order reaction rate constant k) with reactant Y , generating Z and regenerating O , which also diffuses through the film with diffusion coefficient D_o . In contrast to O and R , Y and Z are freely diffusing (diffusion constant D_Y and D_Z respectively) both within the film and in the surrounding solution. Due to the rapid rotation of the external solution, the concentration of Y at the film/solution interface remains constant at its total value (Y_{Tot}), which is equal to its concentration in the surrounding solution, which is also uniform. Within the film, however, the concentration of Y can vary with position.

Modeling Equations. The modeling equations consist of material balances on O and Y within the redox film (equations S1 and S2 respectively), resulting in a system of two partial differential equations (PDEs) that must be solved simultaneously for the time and space dependent concentration profiles of O and Y .

$$\frac{\partial C_O}{\partial t} = D_o \frac{\partial^2 C_O}{\partial x^2} + k(O_{Tot} - C_O)(C_Y) \quad (S1)$$

$$\frac{\partial C_Y}{\partial t} = D_Y \frac{\partial^2 C_Y}{\partial x^2} - k(Y_{Tot} - C_O)(C_Y) \quad (S2)$$

The current is calculated based on the time dependent concentration gradient of O at the electrode surface (equation S3).

$$I = -nFAD_o \left(\frac{\partial C_o}{\partial x} \right)_{x=0} \quad (\text{S3})$$

Initial and Boundary Conditions. At the initial condition, C_O and C_Y are equal to their total concentrations (O_{Tot} and Y_{Tot}), respectively. The electron transfer at the electrode surface is rapid and is modeled according to Butler-Volmer kinetics; with respect to Y , the electrode is inert and impermeable. At the film/solution boundary, the flux of O and R out of the film is zero since it is confined, and the concentration of O is equal to Y_{Tot} because of perfect mixing in the external solution as a result of the high rotation speed.

Numerical Solution of the PDE system. Before solving numerically, the time, space, and concentration variables were all scaled with respect to their maximum values. The system was numerically solved by means of the method of lines, in which the space variable was discretized, resulting in an equivalent system of simultaneous ordinary differential equations (ODEs). The set of finite difference ODEs were individually derived by material balances within the discretized control volumes (finite volume method). The ODE system was solved numerically using the Julia programming language, by making use of the LSODA solver within the DifferentialEquations.jl package.

Fig. S10: Electrochemical O_2 reduction at bare GCE and polymer coated GCE. (a) CVs of viologen-modified hydrogel films on GCE (orange) and bare GCE (black) under 5% O_2 in Ar. Both measurements were performed at 298 K in phosphate buffer (0.1 M, pH 7) at a scan rate of

10 mV s⁻¹ with a rotation rate of 2000 rpm. Surface coverage of the polymer on the electrode is 0.3 mg cm⁻². (b) Simulation results for polymer coated electrodes based on the kinetic scheme in Fig. S9 with the following parameter values: $A = 0.071 \text{ cm}^2$, $O_{Tot} = 60 \text{ mM}$, $Y_{Tot} = 0.05 \text{ mM}$, $k = 1 \times 10^6 \text{ M}^{-1} \text{ s}^{-1}$, $E^0 = -0.210 \text{ V}$, scan rate = 0.10 V s⁻¹, $D_O = D_R = 1.0 \times 10^{-9} \text{ cm}^2 \text{ s}^{-1}$ and $D_Y = D_Z = 1.0 \times 10^{-5} \text{ cm}^2 \text{ s}^{-1}$.

(4) Can KI be added onto the electrode film directly? Is there any way to minimize the loading of KI?

Having iodide only in the film would be advantageous because it would not be able to reach the counter electrode (where it gets oxidized to I₂). But this is not relevant for practical applications since in biofuel cell the H₂ oxidation electrode is separated from the O₂ reducing electrode by a proton exchange membrane. Therefore, it is not necessary to restrain KI to the film or minimize its loading.

(5) More data should be provided to evaluate the performance of the H₂/O₂ biofuel cell when KI was used to protect the viologen polymer film. How about the influence of the high operating temperature?

We performed these experiments and made the additions to the main text and SI as follow:

“The applicability of the iodide containing electrolyte in H₂/O₂ biofuel cells was tested by coupling the hydrogenase modified anode to an O₂ reducing cathode modified with the enzyme bilirubin oxidase in a two compartment cell. The anode compartment was filled with phosphate buffer containing iodide and the cathode compartment with phosphate buffer only to avoid iodide oxidation. The fuel cell performance tests yielded current densities of $117.6 \pm 31.6 \mu\text{A cm}^{-2}$, power densities of $5.8 \pm 1.2 \mu\text{W}$ and an open circuit voltage of $1.0 \pm 0.05 \text{ V}$ (Fig. 5b). These performances are comparable with the ones obtained from fuel cells that do not contain iodide and thus demonstrate the absence of detrimental effects of its use as electrolyte in the anodic compartment.”

Fig. 5: Turnover stability of viologen-modified films containing hydrogenase. (a) The catalytic current densities for H₂ oxidation in presence of 5% O₂ are normalized (j_{norm}) by their maximum values. The measurements are performed in phosphate buffer (0.1 M, pH 7) in the absence or in the presence of KI (0.1 M). Replicates of measurements over extended time periods under the same conditions are given in **Fig. S12**. Measurement conditions: The potential was held at 0.21 V vs SHE. The rotation rate was 2000 rpm. (b) Current density (black trace) and power density (red trace) versus cell voltage from fuel cell measurements conducted in a two compartment cell with a glass frit as separator. The anode modified with a film of the viologen polymer containing the hydrogenase was placed in the compartment containing phosphate buffer (0.1 M, pH 7) and KI (0.1 M) purged with 100% H₂. The cathode modified with bilirubin oxidase for O₂ reduction was placed in the compartment containing phosphate buffer (0.1 M, pH 7) purged with 100% O₂. Replicates of the fuel cell measurements under the same conditions are given in **Fig. S13**. The polymer surface coverage for all hydrogenase modified electrodes is 2.26 mg cm⁻². All measurements were performed at 298 K.

The H₂-oxidizing anode was also tested at 40 °C:

“By contrast, in presence of I⁻ the loss in catalytic current was only 9% in the same time period and prolonged measurements showed a half-life of 7.3 ± 1.9 days) (**Fig. S12b, d**), which is also reached at elevated temperatures of 40°C (**Fig. S12e** half-life: 6.3 ± 1.4 days).”

Fig. S12: Turnover stability of viologen-modified films containing hydrogenase. Catalytic current (bottom) from the polymer-hydrogenase electrode (**a**), (**c**) in the absence and (**b**), (**d**), (**e**) in the presence of KI (0.1M). The measurements (**a**), (**b**), (**c**) and (**d**) were conducted at 298 K and (**e**) at 333K under aerobic conditions (5% O₂ in Ar, the concentration of O₂ in the electrolyte is given as the upper). All measurements were performed in phosphate buffer (0.1 M, pH 7) at an applied potential of 0.21 V vs SHE with a rotation rate of 2000 rpm. Surface coverage of the polymer on the electrode was 2.26 mg cm⁻².

REVIEWERS' COMMENTS:

Reviewer #1 (Remarks to the Author):

This is a really beautiful work. All my comments have been properly addressed and this MS can now be accepted

Reviewer #2 (Remarks to the Author):

This reviewer has no further questions or comments.

Reviewer #3 (Remarks to the Author):

The authors have intensively revised the manuscript and addressed all my concerns. I am satisfied with the current form and recommend an acceptance of this nice work.